# Structure-based nuclear import mechanism of histones H3 and H4 mediated by Kap123

Sojin An[1†], Jungmin Yoon[2†], Hanseong Kim[1], Ji-Joon Song[2], Uhn-soo Cho[1]*

[1]Department of Biological Chemistry, University of Michigan Medical School, Michigan, United States; [2]Structural Biology Laboratory of Epigenetics, Department of Biological Sciences, Graduate school of Nanoscience and Technology (World Class University), KI for the BioCentury, Korea Advanced Institute of Science and Technology, Daejeon, South Korea

**Abstract** Kap123, a major karyopherin protein of budding yeast, recognizes the nuclear localization signals (NLSs) of cytoplasmic histones H3 and H4 and translocates them into the nucleus during DNA replication. Mechanistic questions include H3- and H4-NLS redundancy toward Kap123 and the role of the conserved diacetylation of cytoplasmic H4 (K5ac and K12ac) in Kap123-mediated histone nuclear translocation. Here, we report crystal structures of full-length *Kluyveromyces lactis* Kap123 alone and in complex with H3- and H4-NLSs. Structures reveal the unique feature of Kap123 that possesses two discrete lysine-binding pockets for NLS recognition. Structural comparison illustrates that H3- and H4-NLSs share at least one of two lysine-binding pockets, suggesting that H3- and H4-NLSs are mutually exclusive. Additionally, acetylation of key lysine residues at NLS, particularly H4-NLS diacetylation, weakens the interaction with Kap123. These data support that cytoplasmic histone H4 diacetylation weakens the Kap123-H4-NLS interaction thereby facilitating histone Kap123-H3-dependent H3:H4/Asf1 complex nuclear translocation.

DOI: https://doi.org/10.7554/eLife.30244.001

*For correspondence:
uhnsoo@med.umich.edu

[†]These authors contributed equally to this work

Competing interests: The authors declare that no competing interests exist.

## Introduction

The basic unit of eukaryotic chromatin is the nucleosome, where 146 base pairs of DNA wrap around a histone octamer composed of two copies of core histones H3, H4, H2A, and H2B (*Luger et al., 1997*; *White et al., 2001*). Each core histone contains an unstructured N-terminal tail that possesses NLSs and several known sites for post-translational modifications (PTMs) including acetylation, methylation, phosphorylation and ubiquitylation (*Strahl and Allis, 2000*). These histone tail chemical modifications play important roles in controlling many DNA template-dependent processes, such as gene transcription, DNA replication, DNA repair, and histone deposition and nucleosome assembly (*Strahl and Allis, 2000*; *Berger, 2002*; *Cosgrove and Wolberger, 2005*; *Jenuwein and Allis, 2001*; *Krebs, 2007*; *Marmorstein, 2001*). The entirety of the genomic DNA, including the number of nucleosomes, is duplicated during S phase. While DNA polymerases replicate the genomic DNA, pre-existing nucleosomes undergo disassembly and reassembly processes from one parental chromatid to one of two sister chromatids (*Ransom et al., 2010*). Meanwhile, the same amount of nascent histones are synthesized at the cytosol and imported into the nucleus to meet the demand for nucleosome shortage (*Franco et al., 2005*; *Verreault, 2000*).

Histones H3 and H4 cytoplasmic assembly and nuclear translocation are relatively well illustrated (*Campos et al., 2010*; *Keck and Pemberton, 2012*). Several heat-shock proteins initially assist in histone H3 and H4 protein folding. Asf1, a H3/H4-specific histone chaperone, subsequently associates

with the histones and forms a heterotrimeric H3:H4/Asf1 complex (*Campos et al., 2010*; *Keck and Pemberton, 2012*; *Tyler et al., 1999*; *Natsume et al., 2007*; *English et al., 2006*). Asf1 shields histones H3 and H4 from undesired non-specific interactions and safely delivers them into the nucleus for de novo nucleosome assembly. Newly synthesized histones also undergo a series of PTMs at the cytoplasm. Soon after their synthesis, histones H3 and H4 are prone to be acetylated mainly at the N-terminal tail, where their NLSs are located. The most evolutionarily conserved PTM identified to date is the diacetylation of histone H4 at K5 and K12 (H4 K5ac and K12ac) (*Sobel et al., 1994*; *Sobel et al., 1995*; *Loyola et al., 2006*; *Jasencakova et al., 2010*). Although highly conserved among eukaryotes, histone H4 K5 and K12 cytoplasmic diacetylation does not appear to be essential for cell survival (*Ai and Parthun, 2004*; *Barman et al., 2006*). Additionally, in fungi, the diacetylation appears to be dispensable for nucleosome deposition, suggesting a potential role in the nuclear translocation (*Ma et al., 1998*).

Macromolecules larger than 40 kDa cannot pass through the nuclear pore complex by simple diffusion and require the assistance of active transporters called karyopherins (Kaps) (*Stewart, 2007*). The family of Kap proteins recognizes either NLS or the nuclear export signal in order to import or export cargo proteins, respectively (*Baake et al., 2001*; *Mühlhäusser et al., 2001*). To date, nineteen human Kaps and fourteen budding yeast Kaps have been identified (*Görlich and Kutay, 1999*). Among them, eleven human and ten budding yeast Kaps are known to mediate the import of cargo proteins to the nucleus and the rest are responsible for either unidirectional export or the bidirectional import/export process of cargo proteins (*Chook and Süel, 2011*). Kap proteins share the common structural motif of helical tandem HEAT (**H**untingtin, elongation factor 3 (**E**F3), protein phosphatase 2A (PP2**A**), and the yeast kinase **T**OR1) repeats with a right-handed superhelical solenoid structure (*Stewart, 2007*; *Mosammaparast and Pemberton, 2004*; *Xu et al., 2010*). The NLS is a structurally disordered region of cargo proteins that is composed of overall positively charged residues with limited sequence homology (*Lee et al., 2006*). Kap proteins mostly recognize NLSs through their inner concave surface (*Xu et al., 2010*; *Lee et al., 2006*; *Marfori et al., 2011*; *Kobayashi and Matsuura, 2013*). The lack of sequence conservation among NLSs allows a small number of Kap proteins to translocate a large number of nuclear proteins into the nucleus, which means that a single Kap protein can translocate many substrates (*Chook and Süel, 2011*). Conversely, a single cargo protein can be imported via several Kap proteins. Kap123, the most abundant karyopherin in budding yeast, functions as a primary transporter for the H3:H4/Asf1 complex, whereas Kap121 acts as a secondary transporter (*Mosammaparast et al., 2002*).

Although the nuclear import of histones is an important biological process in DNA replication, the detailed mechanism by which Kap proteins recognize and regulate histone-NLSs remains elusive. It is still not known how Kap123 recognizes the H3:H4/Asf1 complex. As both histones H3 and H4 are redundant and possess NLS peptides at their N-termini (*Mosammaparast et al., 2002*; *Blackwell et al., 2007*), Kap123 either recognizes both H3- and H4-NLSs simultaneously or associates with only one of them preferentially. It is also not clear whether and how PTMs, particularly histone H3 and H4 acetylation, impact H3:H4/Asf1 complex nuclear transport. Mutations of key H3- and H4-NLS lysine residues to glutamines, an acetylation mimic of the lysine residue, impaired H3-NLS- and H4-NLS-GFP reporter translocation (*Blackwell et al., 2007*). This suggests that acetylation of NLSs interferes with rather than promotes histone H3 and H4 nuclear translocation.

Here, we report crystal structures of full-length *Kluyveromyces lactis* (*Kl*) Kap123 alone and in complex with NLS peptides of H3 and H4. We identify that there are two spatially separated lysine-binding pockets within Kap123 that specifically interact with H3- and H4-NLSs. The expected NLS consensus sequence for Kap123 interaction, derived from the Kap123-H3-NLS structure, is $-X_{SH}-K-X_{SH}-(X)_{6\ or\ more}-K-$ ($X_{SH}$: small hydrophobic amino acid, X: any amino acid). Structural comparison of Kap123-H3-NLS and Kap123-H4-NLS demonstrates that H3- and H4-NLSs share at least one of the lysine-binding pockets within Kap123, indicating that H3- and H4-NLSs are mutually exclusive with respect to Kap123 association. The structures also suggest that the acetylation of key lysine residues weakens the Kap123-NLS interaction by losing electrostatic interactions (*Blackwell et al., 2007*). This indicates that the cytosolic acetylation of H3- and H4-NLSs, such as the conserved diacetylation of H4 K5 and K12, may interrupt the Kap123-dependent recognition of H3-/H4-NLSs during nuclear translocation. Based on structural observations and subsequent mutational analysis, we propose that the diacetylation of histone H4 (H4 K5ac and K12ac) may exclude H4-NLS from Kap123, thereby

leading to the histone H3-dependent nuclear translocation of the H3:H4/Asf1 complex mediated by Kap123.

## Results

### Overall structure of full-length *Kl* Kap123

The crystal structure of full length *Kl* Kap123 was determined using the single-wavelength anomalous diffraction (SAD) method and refined at 2.35 Å resolution with a crystallographic R value of 20.99% and a free R value of 23.49% (*Supplementary file 1*). Crystals contain two copies of full-length *Kl* Kap123 per asymmetric unit. The refined structure displays 24 tandem HEAT repeats with a right-handed superhelical solenoid structure (*Figure 1a*). The first HEAT repeat (residues 1–39), an intra-loop of repeat 8 (residues 325–329), repeat 15 (residues 631–656) and repeat 18 (residues 818–828) are disordered in the final structure (*Figure 1a*). The overall architecture of *Kl* Kap123 is similar to other known karyopherin structures (*Xu et al., 2010*; *Lee et al., 2006*; *Marfori et al., 2011*; *Kobayashi and Matsuura, 2013*), excepting repeat 23 (*Figure 1b*, *Figure 1—figure supplement 1*, and *Figure 2—figure supplement 1*). The unique extra-long helix of repeat 23 is conserved in budding yeasts, such as *Saccharomyces cerevisiae* and *K. lactis*, and establishes an intra-molecular interaction with repeats 12–14 using electrostatic and hydrogen bond interactions. This interaction may further stabilize the superhelical architecture of *Kl* Kap123 (*Figure 1b* and *Figure 3—figure supplement 1*).

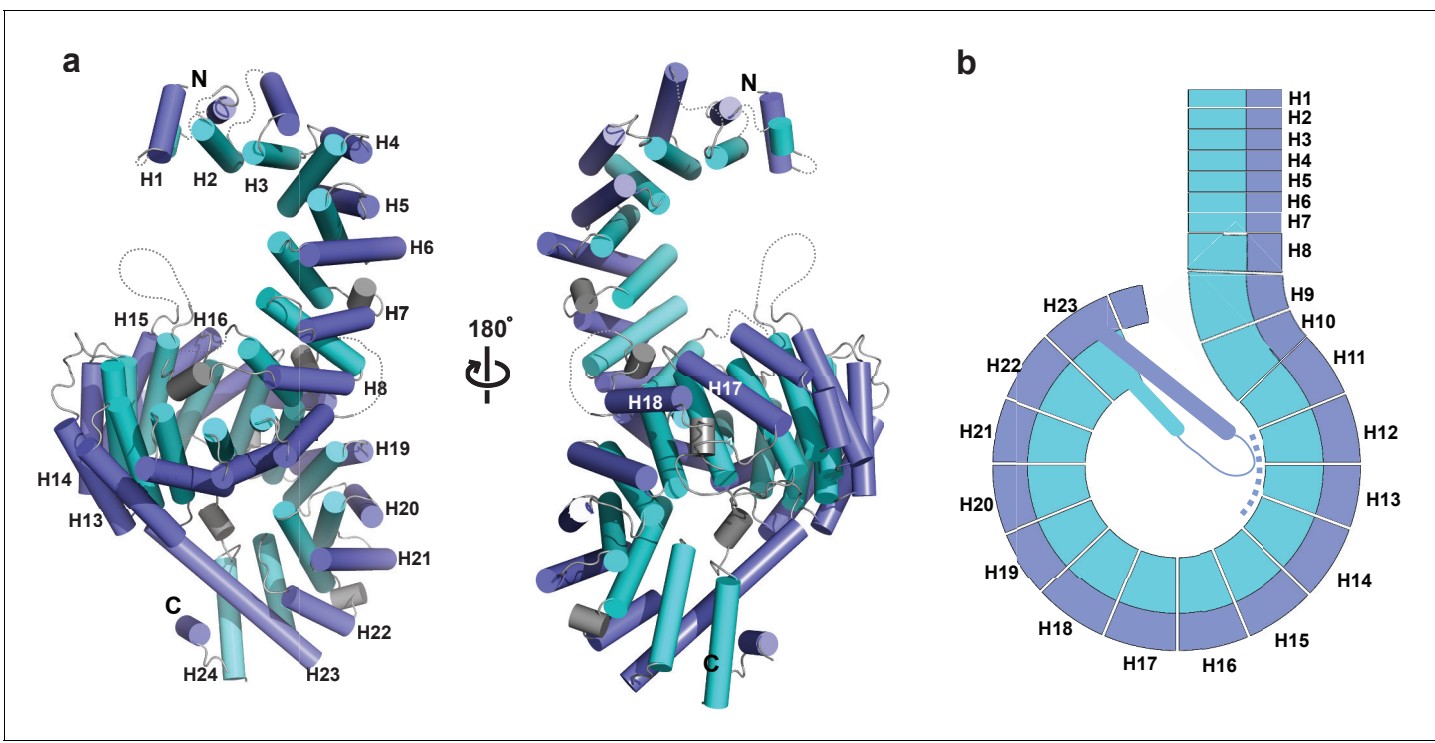

**Figure 1.** Crystal structure of full-length *Kluyveromyces lactis* (*Kl*) Kap123. (a) The apo structure of full-length *Kl* Kap123 with two lateral views (180° rotation). The inner (cyan) and outer (blue) helices make up the 24 Kap123 HEAT repeats (H1–H24). The HEAT repeats form a right-handed superhelical solenoid structure with several linker regions (gray). (b) Schematic view of full-length *Kl* Kap123. The extra-long helix of repeat 23 forms an intramolecular interaction with repeats 12–14. The illustration incorporated in all figures was generated using PYMOL (Delano Scientific, LLC).
DOI: https://doi.org/10.7554/eLife.30244.002

The following figure supplement is available for figure 1:

**Figure supplement 1.** Structural comparison of full-length *S. cerevisiae* Kap121 and full-length *K. lactis* Kap123.
DOI: https://doi.org/10.7554/eLife.30244.003

## *Kl* Kap123 recognizes H3$_{1-28}$-NLS via two lysine-binding pockets

Earlier studies identified that both histones H3 and H4 contain NLSs at their N-terminal unstructured regions and are functionally redundant (*Mosammaparast et al., 2002*; *Blackwell et al., 2007*). In budding yeast, residues 1–28 of histone H3 (H3$_{1-28}$-NLS) and residues 1–34 of histone H4 (H4$_{1-34}$-NLS) were respectively designated as histone H3 and H4 NLSs (*Mosammaparast et al., 2002*; *Blackwell et al., 2007*).

To examine how Kap123 recognizes H3-NLS, we determined the full-length *Kl* Kap123 crystal structure in the presence of a histone H3$_{1-28}$ peptide ($_1$ARTKQTARKSTGGKA PRKQLASKAARK$_{28}$) at 2.7 Å resolution (*Supplementary file 1* and *Figure 2a*). No major structural change of Kap123 was observed upon the H3 peptide binding with Cα root-mean-square deviations (RMSDs) of 0.4 Å.

The co-crystal structure of the Kap123-H3$_{1-28}$-NLS complex displayed extra electron density belonging to the histone H3 peptide (*Figure 2—figure supplement 2*). Although we used the histone H3$_{1-28}$ peptide for crystallization, only residues 12–17 and 21–26 of histone H3 ($_{12}$GG$\underline{K}$APR$_{17}$ and $_{21}$AS$\underline{K}$AAR$_{26}$) were clearly visible, indicating that H3 peptide residues 1–11, 18–20 and 27–28 were disordered in the crystal structure and not critical for H3-NLS recognition by Kap123 (*Figure 2a,e*). The determined structure revealed that the H3$_{1-28}$-NLS peptide interacts with Kap123 through two deep lysine-binding pockets separately located at the Kap123 inner concave surface (*Figure 2a,b*). The binding pattern of H3$_{1-28}$-NLS toward Kap123 is unique and clearly distinguishable from any known karyopherin-NLS structures (*Xu et al., 2010*; *Lee et al., 2006*; *Marfori et al., 2011*; *Kobayashi and Matsuura, 2013*). Most NLSs within Kap-NLS structures recognize Kap proteins using their extended conformation that continuously or sequentially interacts with the C-terminal Kap protein inner surface region (*Xu et al., 2010*; *Lee et al., 2006*; *Marfori et al., 2011*; *Kobayashi and Matsuura, 2013*). K14 and K23 of H3$_{1-28}$-NLS independently interact with two distinct and distal lysine-binding pockets, demonstrating that H3$_{1-28}$-NLS recognizes Kap123 in the bipartite binding mode (*Figure 2b,e*). Kap123 does not recognize the specific sequence of the middle region (residues 15–22) of H3$_{1-28}$-NLS (*Figure 2b,e*). Only several peptide backbone interactions via hydrogen bond interactions are observed in-between Kap123 and H3$_{1-28}$-NLS except key lysine residues (Fb). Even though bipartite binding has been observed in classical NLS models (*Fontes et al., 2000*), the classical NLS-binding pattern is more continuous and binds on the surface, rather than via the binding pocket. Therefore, the Kap123-H3$_{1-28}$-NLS complex crystal structure demonstrates that *Kl* Kap123 recognizes H3$_{1-28}$-NLS peptides in a unique bipartite manner using two distally positioned lysine-binding pockets.

## The first and second lysine-binding pockets of *Kl* Kap123 recognize K14 and K23 of H3$_{1-28}$-NLS, respectively

The first lysine-binding pocket is organized through the inner surface residues of repeats 20–22 of Kap123 (*Figure 2b,c*). Kap123 Y926 (repeat 20) and the stretched aliphatic chain of H3 K14 form a hydrophobic interaction. The negatively charged pocket composed of N980, E1016, and E1017 (repeats 21 and 22) and the positively charged ε-amine group of H3 K14 form several electrostatic and hydrogen bond interactions, which provide specificity toward H3 K14. The peptide backbone of H3$_{1-28}$-NLS is further stabilized by hydrogen bond interactions through E889 (repeat 19), N923 (repeat 20) and R976 (repeat 21) of Kap123 (*Figure 2—figure supplement 3a*). This backbone interaction strongly prefers residues with a small hydrophobic side chain (Ala and Gly) near the key lysine residue, which may provide additional specificity at the first lysine-binding pocket with the consensus sequence of -X$_{SH}$-K-X$_{SH}$- (X$_{SH}$; small hydrophobic amino acid).

The second lysine-binding pocket is established with repeats 11–13 of Kap123 (*Figure 2b,d*). H3 K23 binds to the second lysine-binding pocket through the hydrophobic interaction with F512 (repeat 12) as well as electrostatic interactions with D465 (repeat 11), S505, S509 (repeat 12), and N556 (repeat 13) in a similar manner as that of the first lysine-binding pocket. The peptide backbone interaction of H3$_{1-28}$-NLS near H3 K23 is also observed via hydrogen bond interactions with E469 (repeat 11), R562 (repeat 13), and N601 (repeat 14) of Kap123 (*Figure 2—figure supplement 3b*).

Both Kap123 lysine-binding pockets produce a negatively charged groove to accommodate a lysine residue side chain. The overall architecture of both lysine-binding pockets closely resembles the aromatic cage observed in the PHD finger domain, particularly similar to the H3K4me0 binding motif (*Sanchez and Zhou, 2011*). Lysine residue acetylation abolishes electrostatic interactions

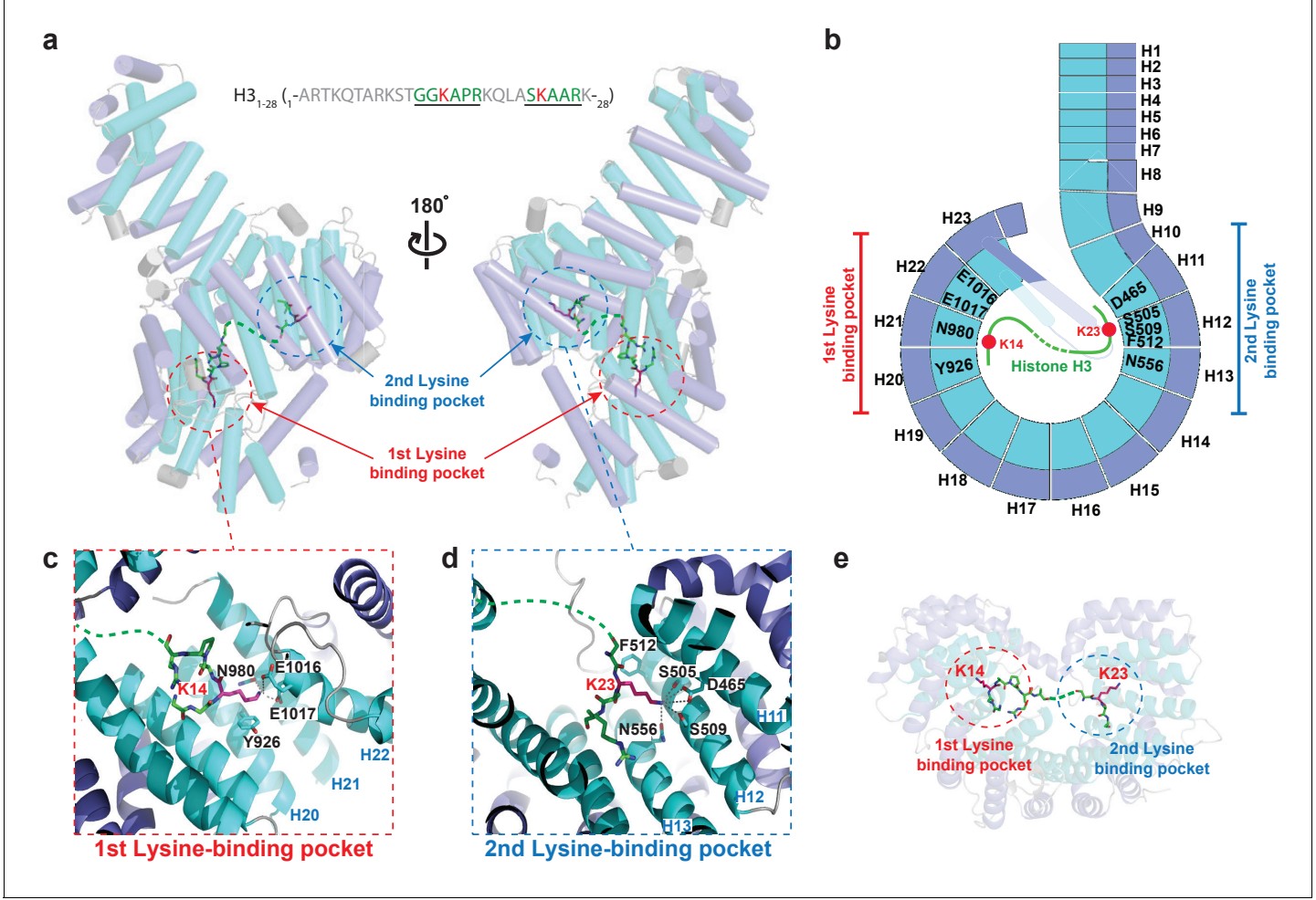

**Figure 2.** Crystal structure of *Kl* Kap123 in complex with H3$_{1-28}$-NLS. (a) Crystal structure of full-length *Kl* Kap123 in the presence of H3$_{1-28}$-NLS (green stick model, $_1$-ARTKQTARKSTGGKAPRKQLASKAARK-$_{28}$) with two lateral views (180° rotation). The two lysine-binding pockets located at the inner curvature of Kap123 are marked with red (first lysine-binding pocket) and blue (second lysine-binding pocket) dashed circles. Residues 12–17 and 21–26 of H3$_{1-28}$-NLS (chain B) are ordered and visible in the structure. The two lysine residues (H3 K14 and K23) that bind to these lysine-binding pockets of Kap123 are colored red. (b) Schematic view of *Kl* Kap123 in complex with H3$_{1-28}$-NLS. The residues and HEAT repeats that participate in organizing two lysine-binding pockets are described. (c) The first lysine-binding pocket of *Kl* Kap123. K14 of H3$_{1-28}$-NLS forms hydrophobic (Y926) and electrostatic/ hydrogen bond (N980, E1016, and E1017) interactions with Kap123 through repeats 20–22. (d) The second lysine-binding pocket of *Kl* Kap123. K23 of H3$_{1-28}$-NLS makes hydrophobic (Y512) and electrostatic/hydrogen bond (D465, S505, S509, and N556) interactions with Kap123 through repeats 11–13. (e) Top view of *Kl* Kap123 in complex with H3$_{1-28}$-NLS. Two lysine-binding sites are distally located and the middle region of H3$_{1-28}$-NLS does not make any specific contacts with Kap123.

DOI: https://doi.org/10.7554/eLife.30244.004

The following figure supplements are available for figure 2:

**Figure supplement 1.** Multiple sequence alignment of Kap121 and Kap123 in budding yeast.
DOI: https://doi.org/10.7554/eLife.30244.005

**Figure supplement 2.** The Composite omit map (1.0 σ contour level) of the Kap123-H3$_{1-28}$-NLS complex.
DOI: https://doi.org/10.7554/eLife.30244.006

**Figure supplement 3.** H3$_{1-28}$-NLS peptide backbone interactions in the Kap123-H3$_{1-28}$-NLS complex.
DOI: https://doi.org/10.7554/eLife.30244.007

**Figure supplement 4.** Structural comparison of Kap123-H3$_{1-28}$-NLS and Kapβ2-H3$_{1-47}$-NLS complexes.
DOI: https://doi.org/10.7554/eLife.30244.008

thereby triggering H3$_{1-28}$-NLS dissociation from Kap123. A previous H3-NLS-GFP reporter assay demonstrated that an acetylation mimic of K14 (K14Q), but not K9Q and K18Q, dramatically reduced GFP reporter nuclear localization, which is in good agreement with our structure (*Blackwell et al., 2007*). The H3-NLS K14R mutation, which maintains the positive charge of the lysine side chain, showed a minor effect with respect to nuclear localization, indicating that the positive charge of Lys14 plays a key role in the nuclear translocation of H3-NLS mediated by Kap123 (*Blackwell et al., 2007*). To avoid any crystallographic artifacts in the Kap123-H3$_{1-28}$-NLS interaction, we mutated residues located in the two lysine-binding pockets of Kap123 as well as key lysine residues at the H3$_{1-28}$-NLS (K14 and K23) and monitored the affinity change by surface plasmon resonance (SPR). Mutations of residues at the first lysine-binding pocket (Y926A, N980A, E1016A and E1017A) and the second lysine-binding pocket (S505A and S509A) substantially reduced H3$_{1-28}$-NLS and Kap123 binding, indicating that the two lysine-binding pockets are indeed involved in H3$_{1-28}$-NLS recognition (*Figure 3a,b,d*). Notably, mutations that interrupt the intra-molecular interaction between the extended helix of repeat 23 and residues at the ridge of repeats 12–14 of Kap123 almost abolished H3$_{1-28}$-NLS peptide binding (*Figure 3c,d* and *Figure 3—figure supplement 1*). Taken together, we demonstrate that Kap123 uses two lysine-binding pockets in order to recognize and accommodate H3$_{1-28}$-NLS.

## *Kl* Kap123 recognizes H4$_{1-34}$-NLS through the second lysine-binding pocket

Similar to H3-NLS, histone H4 also contains a NLS peptide at the N-terminal tail (*Mosammaparast et al., 2002*; *Blackwell et al., 2007*). To investigate how Kap123 recognizes H4-NLS, we also determined the Kap123-H4$_{1-34}$-NLS complex co-crystal structure at 2.82 Å resolution using a H4$_{1-34}$ peptide for co-crystallization ($_1$SGRGKGGKGLGKGGAKRHRKILRDNIQGITKPAI$_{34}$) (*Figure 4a*). The Kap123-H4$_{1-34}$-NLS structure shows clear electron density for residues 13–20 of H4$_{1-34}$-NLS ($_{13}$GGA<u>K</u>RHRK$_{20}$) (*Figure 4—figure supplement 1*). Notably, H4$_{1-34}$-NLS K16 binds to the second lysine-binding pocket of Kap123 in a similar pattern to H3$_{1-28}$-NLS (*Figure 4b,c*). The same residues that participate in H3 K23 recognition (S505, S509, F512 and N556) form the binding pocket to accommodate H4 K16 (*Figure 5c*). Additionally, side chains of R17 and R19 from H4$_{1-34}$-NLS form an electrostatic interaction with E469 (repeat 11) and electrostatic/hydrophobic interactions with E593 (repeat 14) and Y664 (repeat 15) of Kap123, respectively (*Figure 4c*). Conversely, we failed to observe clear H4$_{1-34}$-NLS electron density at the first lysine-binding pocket. Therefore, the crystal structure of the Kap123-H4$_{1-34}$-NLS complex indicates that Kap123 recognizes K16 of H4$_{1-34}$-NLS mainly through the second lysine-binding pocket of Kap123 (*Figure 4d*).

## Acetylation mimic mutations of H3- and H4-NLSs reduce the affinity toward Kap123

Accumulated evidence illustrates that cytosolic histones H3 and H4 are acetylated prior to nuclear import and believed to play a role in the translocation process mediated by Kap123 (*Sobel et al., 1994*; *Sobel et al., 1995*; *Loyola et al., 2006*; *Jasencakova et al., 2010*; *Ma et al., 1998*; *Blackwell et al., 2007*; *Kuo et al., 1996*). However, the histone H3 acetylation pattern is inconsistent among species and loss of the conserved histone H4 diacetylation pattern at K5 and K12 generated by the Hat1 complex did not show any noticeable phenotype upon Hat1 deletion (*Ai and Parthun, 2004*; *Barman et al., 2006*). The structures determined herein indicate that acetylation of key lysine residues may disrupt electrostatic interactions in the pocket, thus inhibiting the Kap123-NLS interaction (*Figures 2c,d and* and *4c*). Accordingly, we introduced mutations in key lysine residues identified from the crystal structures. In agreement with previous reports, acetylation-mimic mutations of H3 K14 and K23 (H3 K14Q and H3 K23Q) as well as alanine substitution (H3 K14A and H3 K23A) decreased the affinity toward Kap123 (*Figure 5a,c*) (*Blackwell et al., 2007*). The double mutation of H3 K14/K23 (K14A/K23A and K14Q/K23Q) further reduced the affinity, demonstrating that H3 K14 and K23 are key residues for Kap123 association and that acetylation disrupts this interaction (*Figure 5a,c*). The H4-NLS K16A or K16Q mutation also reduced the affinity toward Kap123 although the effect was mild probably owing to the additional contacts generated by H4-NLS R17 and R19 (*Figures 4c* and *5b,d*). Notably, H4-NLS$^{K5Q/K12Q}$ diacetylation mimic mutation reduced affinity more significantly than H4-NLS$^{K16A}$ or H4-NLS$^{K16Q}$, suggesting that Kap123 has an additional

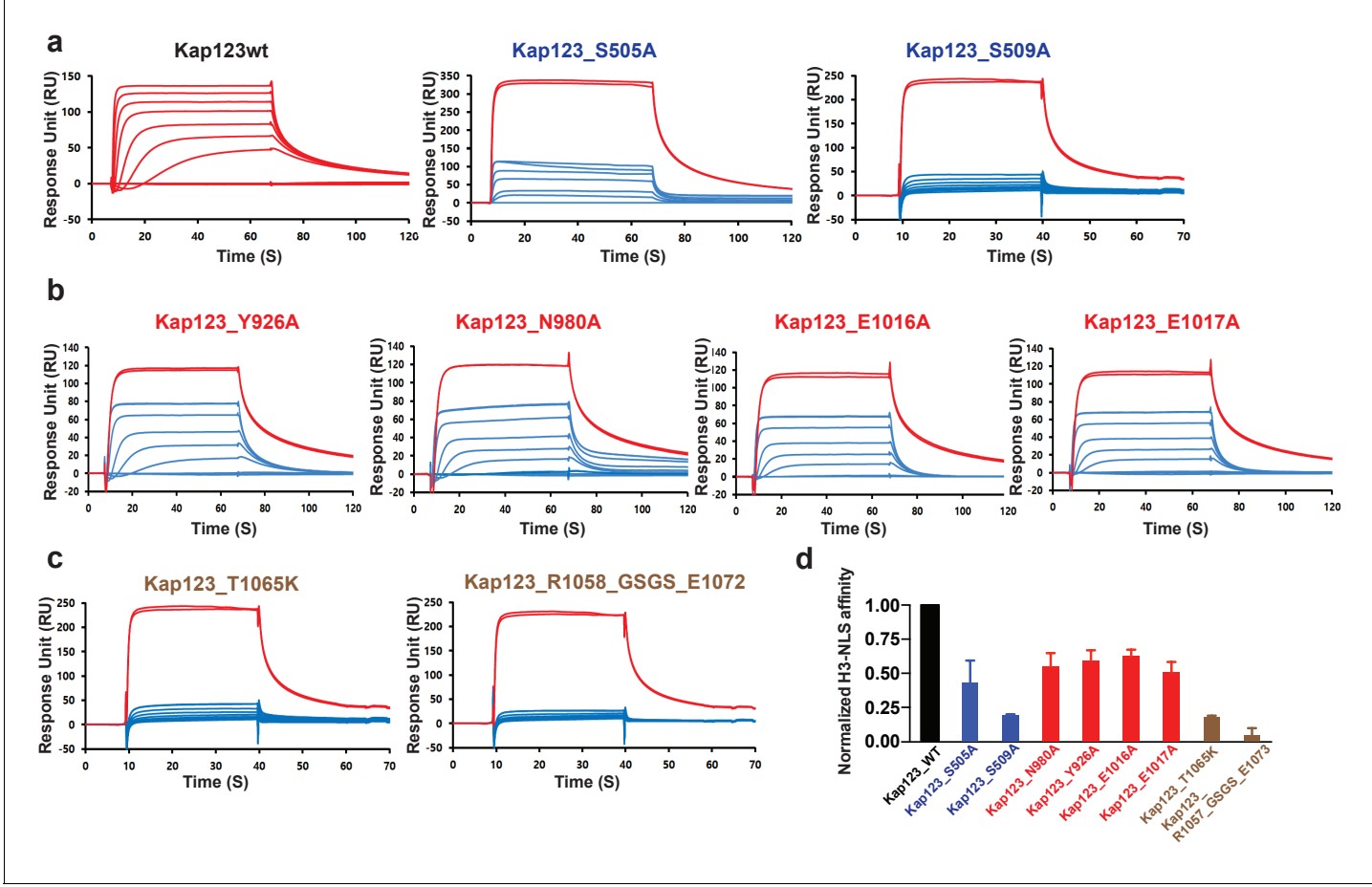

**Figure 3.** Surface plasmon resonance analysis of wild-type and mutants *Kl* Kap123 in the presence of H3$_{1-35}$-NLS. The biacore diagrams of *Kl* Kap123 wild-type and mutants in the presence of H3$_{1-35}$-NLS peptides are listed. (**a**) Streptavidin was immobilized at 400 resonance units (RU) and the C-terminal biotinylated histone H3 peptide (residues 1–35) was captured at 150 RU on a CM5 chip. The wild-type Kap123 was injected with different concentrations (0, 1, 5, 10, 25, 50, 100, 250, and 500 nM) and sensorgrams are colored red. To monitor the affinity of Kap123 mutants at the second lysine-binding pocket (S505A and S509A), the C-terminal biotinylated histone H3 peptide was captured (300 RU) on a streptavidin-immobilized CM5 chip (400 RU). Various concentrations of Kap123_S505A (0, 65, 125, 250, 500, 1000, and 2000 nM) and Kap123_S509A (0, 32, 65, 125, 250, 500, 1000, and 2000 nM) were injected; their sensorgrams are shown in blue. (**b**) Affinity measurement of first lysine-binding pocket mutants of Kap123 (Y926A, N980A, E1016A, and E1017A). Streptavidin (400 RU) and C-terminal biotinylated histone H3 peptide (150 RU) were immobilized on a CM5 chip. The mutant proteins were injected with various concentrations (0, 1, 5, 10, 25, 50, 100, 250, and 500 nM) and sensorgrams are shown in blue. (**c**) Affinity measurement of Kap123 mutants in the extended helix of the repeat 23. Either the point mutation (T1065K) or the deletion mutant (1058-GSGS-1072; residues 1059–1071 are replaced by the GSGS linker) of Kap123 was injected with various concentrations (0, 32, 65, 125, 250, 500, 1000, and 2000 nM for T1065K and 0, 1, 5, 10, 25, 50, 100, 250, and 500 nM for 1058-GSGS-1072). Red-colored sensorgrams are positive controls of wild-type Kap123, which we injected at 2000 nM wild-type Kap123 before and after sample injection (**a–c**). (**d**) Normalized affinity of wild-type and mutant Kap123s toward H3-NLS is shown in the bar graph. The relative affinity of wild-type and mutant Kap123s measured at five different concentrations is used to generate error bars.
DOI: https://doi.org/10.7554/eLife.30244.009

The following figure supplement is available for figure 3:

**Figure supplement 1.** The extended helix of *Kl* Kap123 repeat 23 makes intramolecular interactions with repeats 12–14.
DOI: https://doi.org/10.7554/eLife.30244.010

binding site for H4-NLS in addition to K16. H4 K5 and K12 may play a more significant role in this recognition although the Kap123-H4$_{1-34}$-NLS crystal structure failed to locate the electron density of these two lysines. Therefore, although H4 K16 recognition via the second lysine-binding pocket of Kap123 is more specific, but its contribution to the affinity might not be strong. However, H4 K5 and K12 may contribute more in Kap123-H4-NLS association as indicted in SPR experiments (**Figure 5b, d**). Furthermore, the H3-NLS affinity toward Kap123 is fivefold higher than that of H4-NLS, indicating that H3-NLS is a better substrate for Kap123 association (**Figure 5e**). Consistent with this result,

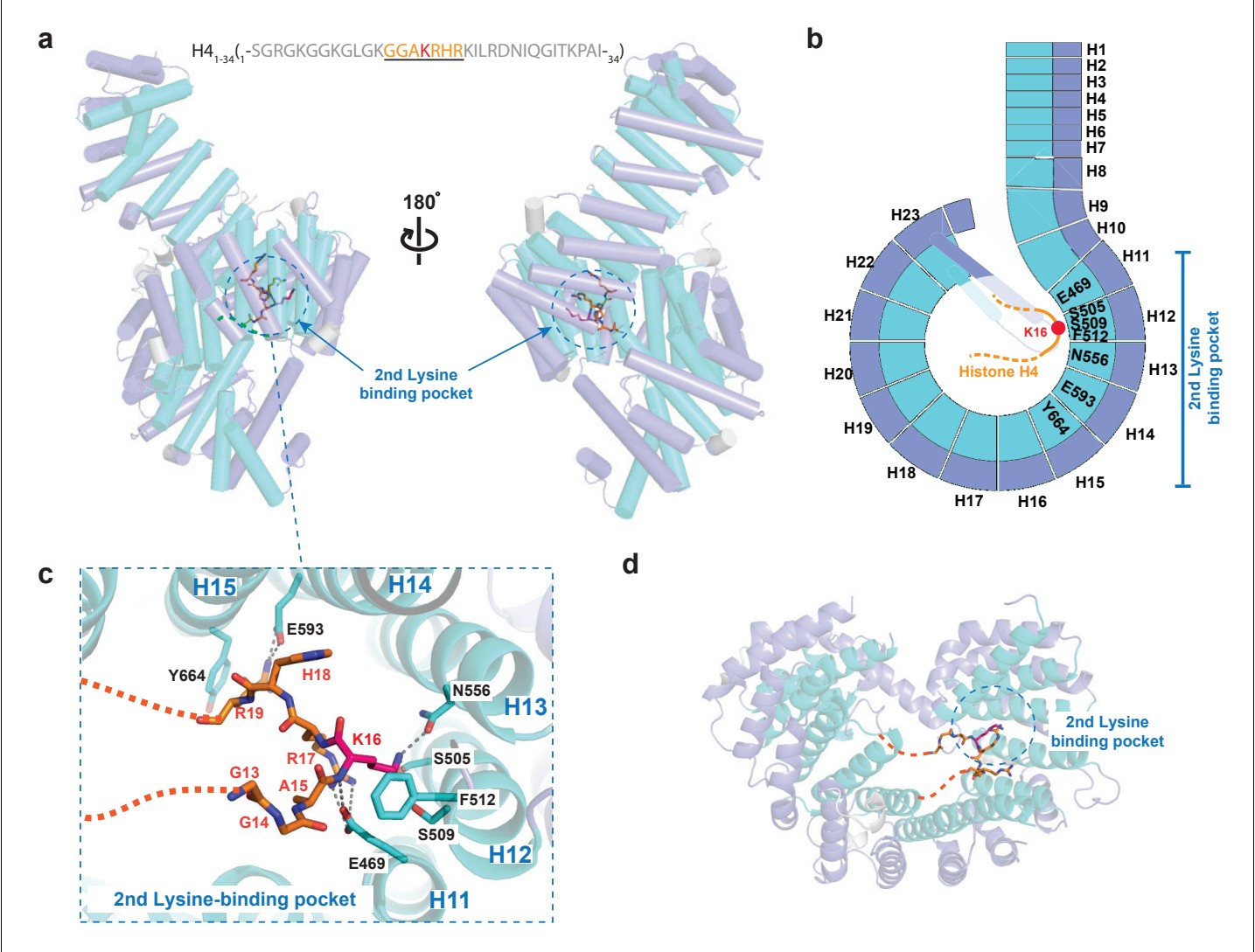

**Figure 4.** Crystal structure of *Kl* Kap123 in complex with H4$_{1-34}$-NLS. (**a**) The crystal structure of full-length *Kl* Kap123 in the presence of H4$_{1-34}$-NLS (orange stick model, $_1$SGRGKGGKGLGKGGAKRHRKILRDNIQGITKPAI$_{34}$) with two lateral views (180° rotation). The second lysine-binding pocket located at the inner curvature of Kap123 is marked as a blue dashed circle. Residues 13–20 of H4$_{1-34}$-NLS are visible in the structure. H4 K16, which binds to the second lysine-binding pocket, is colored red. (**b**) Schematic view of *Kl* Kap123 in complex with H4$_{1-34}$-NLS. Residues of repeats 11–15 that participate in H4$_{1-34}$-NLS recognition are indicated. (**c**) The second lysine-binding pocket of *Kl* Kap123 in H4$_{1-34}$-NLS recognition. K16 of H4$_{1-34}$-NLS forms hydrophobic (F512) and electrostatic/hydrogen bond (S505, S509, and N556) interactions with Kap123 through repeats 12–13. R17 and R19 of H4$_{1-34}$-NLS form additional electrostatic and hydrophobic contacts with E469 (repeat 11) and E593 (repeat 14)/Y664 (repeat 15) of Kap123, respectively. (**d**) Top view of *Kl* Kap123 in complex with H4$_{1-34}$-NLS. Only the second lysine-binding pocket is occupied by K16 of H4$_{1-34}$-NLS.

DOI: https://doi.org/10.7554/eLife.30244.011

The following figure supplement is available for figure 4:

**Figure supplement 1.** The composite omit map (1.0 σ level) of the Kap123-H4$_{1-34}$-NLS complex.

DOI: https://doi.org/10.7554/eLife.30244.012

Soniat *et al.* recently showed a similar result where 11-fold stronger interaction for Kapβ2-H3 than Kapβ2-H4 was observed (*Soniat et al., 2016*). However, we cannot rule out the possibility that H4$_{1-21}$-NLS peptides used for SPR experiments may not fully capture the Kap123 and histone H4 association although we did not observe any additional electron density for H4$_{21-34}$ in the Kap123-H4$_{1-34}$-NLS crystal structure. The former report raised the possibility of the additional interaction between Kap123 and histone H4$_{22-42}$ (*Mosammaparast et al., 2002*).

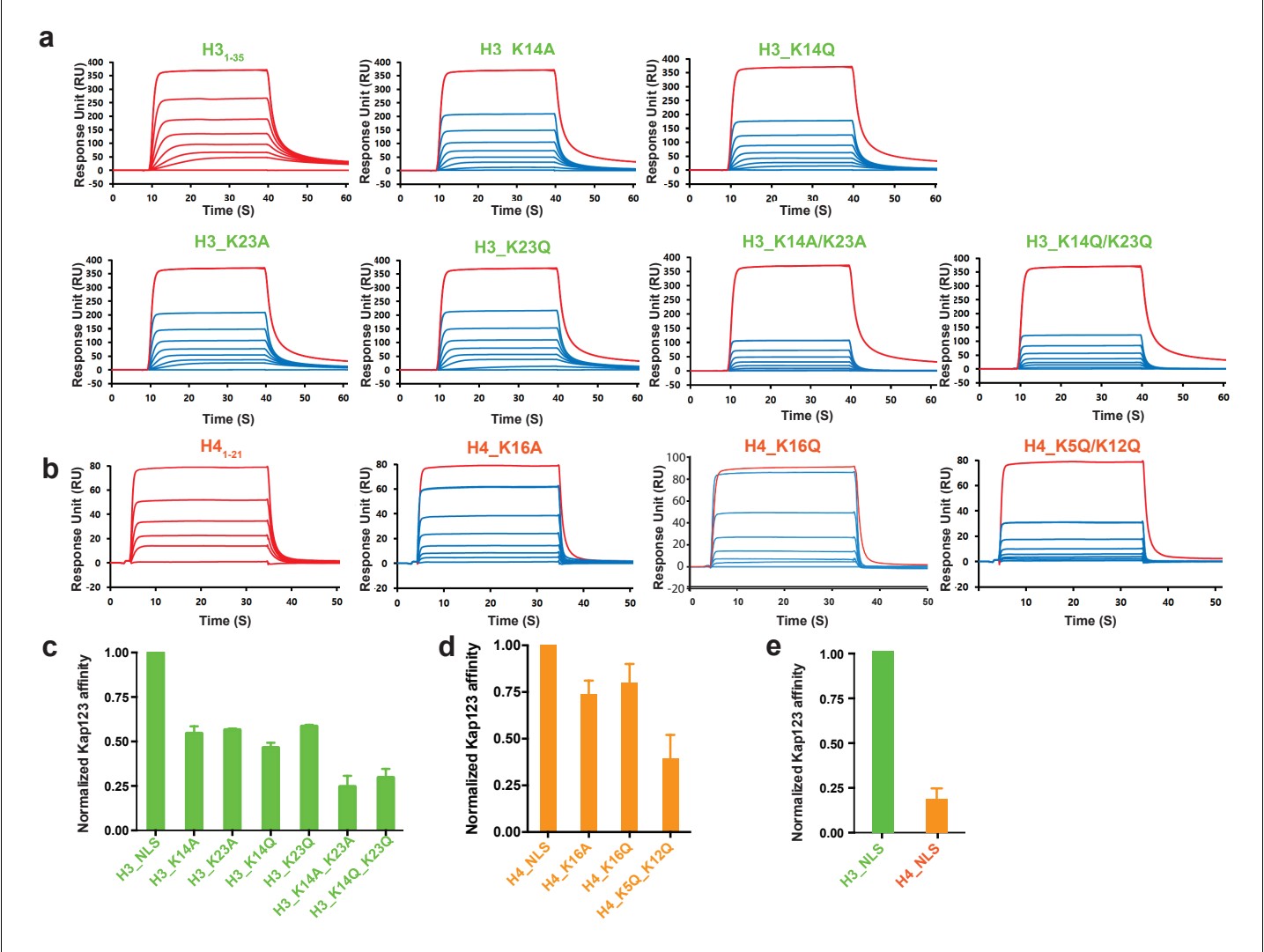

**Figure 5.** Surface plasmon resonance analysis of wild-type *Kl* Kap123 in the presence of wild-type or mutants of H3$_{1-35}$-/H4$_{1-21}$-NLSs. The biacore diagrams (**a**) and the normalized bar graph (**c**) of wild-type Kap123 in the presence of wild-type or mutant H3$_{1-35}$-NLS. The wild-type Kap123 was immobilized on a CM5 chip with 7500 RU and the wild-type or mutant H3$_{1-35}$-NLS peptides (H3_K14A, H3_K14Q, H3_K23A, H3_K23Q, or H3_K14Q/K23Q) were injected at different concentrations (0, 0.07, 0.15, 0.32, 0.65, 1.25, 2.5, and 5 µM). The biacore diagrams (**b**) and the bar graph (**d**) of wild-type Kap123 in the presence of wild-type or mutant H4$_{1-21}$-NLS. The wild-type Kap123 was immobilized on a CM5 chip with 7116 RU and the wild-type or mutant H4$_{1-21}$-NLS peptides (H4_K16A or H4_K5Q/K12Q) were injected at different concentrations (0, 0.32, 0.62, 1.25, 2.5, 5, and 10 µM). The H4K16Q peptide was used at different concentrations (0, 0.7, 1.5, 3, 6, 12, and 25 uM). See details in Materials and method section. Red-colored sensorgrams represent positive controls of wild-type Kap123, which were injected 2000 nM wild-type Kap123 before and after sample injection (**a–b**). The relative binding between wild-type Kap123 and mutant peptides was measured at five different concentrations for generating error bars (**c–e**). (**e**) The normalized affinity bar graph of H3-/H4-NLS toward wild-type Kap123 was derived from (**a**) and (**b**).

DOI: https://doi.org/10.7554/eLife.30244.013

## H3- and H4-NLSs co-compete for Kap123 association

Structural comparison of Kap123-H3$_{1-28}$-NLS and Kap123-H4$_{1-34}$-NLS indicates that at least the second lysine-binding pocket of Kap123 can accommodate both K23 of H3-NLS and K16 of H4-NLS, strongly suggesting that H3- and H4-NLSs compete for Kap123 binding. To validate this observation, we performed a competition assay (*Figure 6*). Pre-incubated Kap123-H3-NLS was challenged by increased amounts of H4-NLS and the remaining amount of H3-NLS monitored after competition. If H3- and H4-NLSs compete for Kap123 association, the increased amount of H4-NLS would compete out H3-NLS from Kap123. As expected, gradually increased H4-NLS successfully competed out

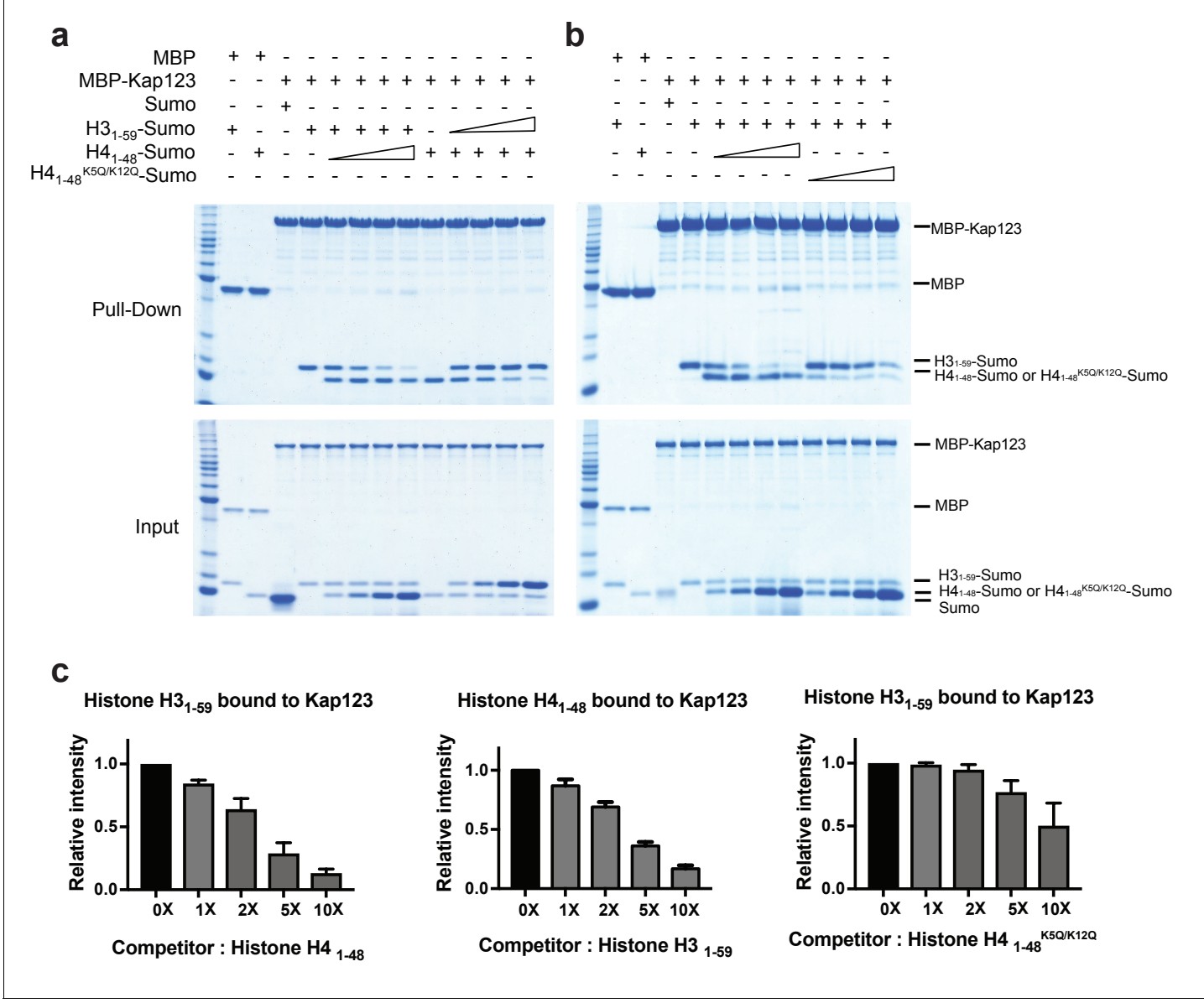

**Figure 6.** Competition assay of H3- and H4-NLSs (wild-type H4 and mutant H4$^{K5Q/K12Q}$) toward Kap123 association. Competition of H3$_{1–59}$- and H4$_{1–48}$-Sumo (**a**) as well as H3$_{1–59}$- and H4$_{1–48}$$^{K5Q/K12Q}$-Sumo (**b**) toward Kap123 binding was monitored by amylose affinity pull-downs using MBP-tagged Kap123. A fixed amount of the pre-assembled complex of MBP-tagged Kap123 and H4$_{1–48}$-Sumo was challenged by an increasing amount of H3$_{1–59}$-Sumo. The same competition assay was done by using pre-incubated MBP-tagged Kap123-H3$_{1–59}$-Sumo and gradually increasing the amount of H4$_{1–48}$-Sumo either with wild-type (**a**) or mutant (**b**), K5Q/K12Q) H4 sequence. (**c–d**) The normalized bar graphs of remaining H3$_{1–59}$- and H4$_{1–48}$-NLS-Sumo after challenged with increased amounts of H4$_{1–48}$- and H3$_{1–59}$-NLS-Sumo (0x, 1x, 2x, 5x, and 10x) toward Kap123, respectively. Either H4$_{1–48}$-NLS-Sumo (**c**) or H4$_{1–48}$$^{K5Q/K12Q}$-NLS-Sumo (**d**) was used for the H3$_{1–59}$-NLS-Sumo competition. Three independent competition assays were carried out and the band intensity was measured to calculate the error bar.

DOI: https://doi.org/10.7554/eLife.30244.014

The following figure supplement is available for figure 6:

**Figure supplement 1.** Competition assay of H3- and H4-NLSs (wild-type H4 and mutant H4$^{K5Q/K8Q/K12Q}$) toward Kap123 association.
DOI: https://doi.org/10.7554/eLife.30244.015

H3-NLS and the same phenomenon occurred in pre-incubated Kap123-H4-NLS with subsequent H3-NLS challenge (*Figure 6a*). This result demonstrated that H3- and H4-NLSs co-compete for Kap123 interaction. The former GFP reporter assay evidenced that the tri-acetylation mimic mutant of H4-NLS$^{K5Q/K8Q/K12Q}$ dramatically reduced the nuclear translocation, probably owing to a reduced affinity toward Kap123 (*Blackwell et al., 2007*). To test whether the tri-acetylation mimic mutant of H4-NLS was still capable of competing with H3-NLS for Kap123 association, we performed the competition assay in a condition where the pre-incubated Kap123-H3-NLS was challenged by an increased amount of H4-NLS$^{K5Q/K8Q/K12Q}$. Unlike the wild-type H4-NLS, H4-NLS$^{K5Q/K8Q/K12Q}$ no longer competed with H3-NLS in Kap123 interaction (*Figure 6—figure supplement 1*). Since evolutionarily conserved histone H4 acetylation happens at K5 and K12, we further performed the competition assay using H4-NLS$^{K5Q/K12Q}$ and we consistently observed loss of competition by using H4-NLS$^{K5Q/K12Q}$ (*Figure 6b,c*). This strongly suggests that the conserved diacetylation pattern of cytoplasmic histone H4 within the H3:H4/Asf1 complex may exclude H4-NLS from Kap123 and allow the Kap123-H3-NLS complex formation during nuclear translocation.

## Discussion

Although the general mechanism by which the family of karyopherin proteins recognizes cargo proteins via NLSs is relatively well understood (*Stewart, 2007*; *Xu et al., 2010*; *Marfori et al., 2011*), two major issues regarding the nuclear transport of histones H3 and H4 have remained elusive. These include: (1) Kap123 recognition of H3- and H4-NLSs. It has not been clear whether one or both NLS signals were required for Kap123 recognition considering the fact that both histones H3 and H4 contain functionally redundant NLSs. (2) Potential role of the acetylation observed in the cytosolic pool of histones H3 and H4. Diacetylation of histone H4 K5 and K12 (H4 K5ac and K12ac) is the most conserved PTM among species; however, the exact biological role of this modification remains ill-defined. In the current study, we focus on addressing these two key questions using crystal structures of full-length *Kl* Kap123 in the presence of either H3$_{1-28}$-NLS or H4$_{1-34}$-NLS and subsequent biochemical and mutational analyses.

### The consensus NLS sequence for *Kl* Kap123 recognition is –X$_{SH}$-K-X$_{SH}$-(X)$_{6 \text{ or more}}$-K-

To gain insights on how Kap123 recognizes the NLSs of histones H3 and H4, we determined crystal structures of a major karyopherin protein in budding yeast, Kap123, in the presence of H3$_{1-28}$-NLS and H4$_{1-34}$-NLS. Determined crystal structures indicate that both H3$_{1-28}$- and H4$_{1-34}$-NLSs interact with Kap123 through either one or both of the lysine binding pockets located in the inner curvature of Kap123. Analysis of how Kap123 interacts with histone H3$_{1-28}$-NLS led to the potential NLS consensus sequence required for Kap123 association. The first lysine-binding pocket (repeats 20–22) of Kap123 captures K14 of H3$_{1-28}$-NLS through both hydrophobic (Y926) and negatively charged (N980, E1016, and E1017) residues (*Figure 2b,c*). The first binding pocket only allows small hydrophobic amino acids (Gly or Ala) as neighboring residues (G13 and A15 in H3$_{1-28}$-NLS) by making several backbone contacts (*Figure 2—figure supplement 3a*). The second lysine-binding pocket (repeats 11–13) of Kap123 recognizes H3 K23 and makes additional backbone interactions with K23 neighboring residues (S22 and A24 of H3$_{1-28}$-NLS) without strong amino acid preference (*Figure 2—figure supplement 3b*). Notably, the two lysine-binding pockets of Kap123 are distally positioned and the distance between the Cα atoms of two bound lysines (H3 K14 and K23) is 26.6 Å, indicating that more than seven residues are required in between two key lysine residues. This results in –X$_{SH}$-K-X$_{SH}$-(X)$_{6 \text{ or more}}$-K- as a consensus sequence for Kap123 recognition.

### H4$_{1-34}$-NLS mainly interacts with Kap123 via the second lysine-binding pocket

A major structural difference between the Kap123-H3$_{1-28}$-NLS and Kap123-H4$_{1-34}$-NLS structures is the missing electron density of the first Kap123 lysine-binding pocket (*Figure 4a,d*). This implies that Kap123 recognizes H4-NLS only through the second-lysine binding pocket. However, we cannot rule out the possibility that K5, K8, and K12 of H4-NLS make additional contacts with Kap123. A previous report showed that the replacement of four lysines at the H4-NLS-GFP reporter (K5, K8, K12 and K16) to alanines led to a defect in nuclear translocation whereas arginine substitution was normal

(*Glowczewski et al., 2004*). Another study showed that the acetylation-mimic triple mutation of K5, K8, and K12 (K5Q/K8Q/K12Q) diminished the nuclear translocation of the H4-NLS-GFP reporter (*Blackwell et al., 2007*). We also observed the diminished affinity of H4-NLS when we introduced the diacetylation-mimic mutations of H4$^{K5Q/K12Q}$ and found that the di-acetylation mimic H4-NLS$^{K5Q/K12Q}$ barely competed with H3-NLS in Kap123 interaction (*Figures 4b,d and* and *6b*). One possible explanation why we could not detect the electron density of H4 K5, K8 and K12 might be that K5 ($_4$-GKG-$_6$), K8 ($_7$-GKG-$_9$), and K12 ($_{11}$-GKG-$_{13}$) of H4-NLS share the same three consecutive amino acid sequence (GKG) and thus equally access the unidentified binding pocket of Kap123. Indeed, earlier studies showed that a single substitution of one of three lysines to glycine or glutamine did not display significant binding defect toward Kap123 indicating that these three lysines are functionally redundant (*Ma et al., 1998*; *Blackwell et al., 2007*). The electron density generated by binding of each lysine may be canceled out and any one of them thereby failed to individually visualize above the noise level. Therefore, although the crystal structure of the Kap123-H4$_{1-34}$-NLS complex only indicates that K16 of H4-NLS mainly associates with the second lysine-binding pocket of Kap123, one of three lysines of histone H4 (K5, K8, or K12) may additionally associate with Kap123. This indicates that H4 K16 weakly but specifically binds to Kap123 while H4 K5, K8, and K12 strongly but less specifically contributes to Kap123 recognition. Further studies are needed to examine the additional binding pocket of H4 K5, K8 and K12 within Kap123.

## The extra-long helix of the Kap123 repeat 23 acts as a molecular ruler

The major structural difference between Kap123 and other known karyopherins is the extra-long helix of the repeat 23 (*Figure 1a,b*, *Figure 1—figure supplement 1*, and *Figure 3—figure supplement 1*). This extended helix can reach to the ridge of repeats 12–14 by forming several charge interactions and hydrogen bond interactions (*Figure 3—figure supplement 1*). Previous small angle x-ray scattering and molecular dynamics simulation results indicated that the overall structures of Kap proteins are highly dynamic in solution (*Forwood et al., 2010*). The lack of structural rigidity may allow Kap proteins to accommodate less conserved NLSs from different cargo proteins. However, the dynamic behavior of Kap123 is thought to be restricted owing to intramolecular interactions between repeats 12 and 14 and the extended helix of the repeat 23. This extended helix restricts the distance in-between two lysine-binding pockets. Indeed, disrupting this intramolecular interaction by either the point mutation of T1065K or the replacement of residues 1059–1071 to a GSGS linker in the extended helix of repeat 23 (R1058_GSGS_E1072) dramatically reduced the H3-NLS affinity toward Kap123 (*Figure 3c,d* and *Figure 3—figure supplement 1*). Therefore, the intramolecular interaction of the repeat 23 at Kap123 may play an important role in H3-NLS recognition of Kap123.

## Association of H3-NLS with karyopherin-β2 (Kapβ2) may compete with other PY-NLSs

While we were preparing the manuscript, the crystal structure of the Kapβ2-H3-NLS complex was released (*Soniat and Chook, 2016*). Kapβ2 is an importin that translocates many cargo proteins into the nucleus by recognizing proline-tyrosine nuclear localization signals (PY-NLSs) of substrates. Although Kapβ2 is not a major importin for core histones (histones H3, H4, H2A and H2B) and histones H3 and H4 do not contain PY-NLSs, Kapβ2 has been shown to recognize H3- and H4-NLSs (*Soniat et al., 2016*). The structural alignment of Kap123-H3$_{1-28}$-NLS and Kapβ2-H3$_{1-47}$-NLS complexes revealed that the binding pattern of H3-NLS is quite distinct (*Figure 2—figure supplement 4a,b*). This strongly suggests that H3-NLS adopts different conformations in order to associate with different Kap proteins. In the Kapβ2-H3$_{1-47}$-NLS structure, H3$_{1-47}$-NLS associates with the inner concave surface of the C-terminal half of Kapβ2 (*Soniat and Chook, 2016*). Residues 11–19 of H3-NLS continuously bind on the inner surface of Kapβ2 and residues 20–27 form a 2-turn α-helix in order to make additional contacts with Kapβ2 (*Figure 2—figure supplement 4b*). Therefore, the H3$_{1-47}$-NLS binding pattern to Kapβ2 apparently differs from the way that Kap123 recognizes H3$_{1-28}$-NLS (*Figure 2—figure supplement 4b*). Another notable observation is that the binding region of H3$_{1-47}$-NLS within Kapβ2 overlaps with that of other known PY-NLSs, which may explain why Kap123 and its human homologue Importin-4 remain dominant transporters for H3- and H4-NLSs in vivo. Although the Kapβ2-H3$_{1-47}$-NLS interaction is strong ($K_D$ of 77.1 nM), H3-NLS needs to compete with other

PY-NLSs in order to associate with Kapβ2, which may limit Kapβ2 access to H3-NLSs in vivo (*Soniat et al., 2016*). The unique bipartite binding mode observed in Kap123 may thus allow selective association with H3-NLS but not other NLSs, such as PY-NLSs.

## The two lysine-binding pockets of Kap123 are not conserved in Kap121

In budding yeast, Kap123 and Kap121 demonstrate functional overlap in importing cargo proteins (*Chook and Süel, 2011*; *Mosammaparast et al., 2002*; *Timney et al., 2006*). Kap121 acts as a secondary transporter for many Kap123 cargoes including the H3:H4/Asf1 complex. Nuclear import of histones H3 and H4 was not disturbed in Kap123Δ cells probably owing to the presence of Kap121 (*Mosammaparast et al., 2002*). Many Kap121 cargoes also use Kap123 as a secondary transporter (*Chook and Süel, 2011*). The crystal structure of *S. cerevisiae* Kap121 has been determined and the overall shape of Kap121 resembles that of *Kl* Kap123 (*Figure 1—figure supplement 1*) (*Kobayashi and Matsuura, 2013*). However, Kap121 and Kap123 also contain unique structural features, such as the H15 insert of Kap121 and the H23 insert of Kap123 (*Figure 1—figure supplement 1*). Particularly, the extended repeat 23 helix of Kap123, whose mutation dramatically reduces the binding ability toward H3-NLS, is not conserved in Kap121 (*Figure 3c,d* and *Figure 2—figure supplement 1*). Moreover, most of the key Kap123 residues important for histones H3 and H4 recognition are well conserved in both *S. cerevisiae* and *K. lactis* Kap123, but not in Kap121 (*Figure 2—figure supplement 1*). Taken together, this indicates that the recognition pattern of Kap121 toward histones H3 and H4 may differ from that of Kap123. The recognition mechanism of Kap121 toward H3- and H4-NLSs may more closely resemble that of Kapβ2 (*Kobayashi and Matsuura, 2013*; *Soniat and Chook, 2016*).

## Diacetylation of H4-NLS may allow Kap123-H3-dependent nuclear translocation of the H3:H4/Asf1 complex

N-terminal tails of both histones H3 and H4 undergo acetylation prior to nuclear translocation; moreover, acetylation is thought to be involved in facilitating the process of karyopherin-dependent nuclear import (*Sobel et al., 1994*; *Sobel et al., 1995*; *Loyola et al., 2006*; *Jasencakova et al., 2010*; *Ma et al., 1998*; *Blackwell et al., 2007*; *Kuo et al., 1996*). Indeed, former reports showed that diacetylation of histone H4 K5 and K12 promotes the nuclear transportation in *P. polycephalum* (*Ejlassi-Lassallette et al., 2011*) and in human (*Alvarez et al., 2011*). However, acetylation or acetylation mimics of lysine residues impaired H3- or H4-NLS-GFP reporter translocation, raising a question regarding the potential role of acetylation in nuclear transport (*Blackwell et al., 2007*; *Glowczewski et al., 2004*). Crystal structures of H3$_{1-28}$-NLS-Kap123 and H4$_{1-34}$-NLS-Kap123 provide further supporting evidence demonstrating that the acetylation of key lysine residues interrupts the Kap123 and NLS interaction by abolishing charge-charge interactions (*Figures 2c,d and* and *4c*). Therefore, we hypothesize that the potential role of either H3- or H4-NLS acetylation is to exclude the binding of NLS toward Kap123 (*Figure 7*). Particularly, H4-NLS diacetylation at K5 and K12 will promote H3-NLS-dependent Kap123 association of the H3:H4/Asf1 complex, thereby facilitating H3-NLS-dependent nuclear translocation during S phase. This may provide selective translocation of histone H3 variants depending on their acetylation status by excluding commonly shared histone H4 from Kap123. In accordance with this speculation, different levels of K14 acetylation were observed in H3.1 (7%) and H3.3 (20%) (*Loyola et al., 2006*), although the K5 and K12 diacetylation pattern on histone H4 was preserved in H3.1, H3.3, and CENP-A pre-deposition complexes (*Loyola et al., 2006*; *Bailey et al., 2016*). Considering the fact that the large amounts of histones translocate during DNA replication, this minor difference may have significant difference in nuclear translocation preference. Follow-up studies are necessary to test this possibility. This observation also suggests that acetylation of H3-NLS, particularly at K14 and K23, may be a part of the regulation mechanism of histone nuclear import. Several groups previously observed H3 K14 acetylation in cytoplasmic histones H3 and H4 (*Loyola et al., 2006*; *Kuo et al., 1996*; *Bailey et al., 2016*).

In the current study, we aimed to address two major issues associated with Kap123-dependent nuclear translocation of histones H3 and H4: (1) Kap123 recognition of the nuclear localization signals of histones H3 and H4; and (2) the role of post-translational modifications, particularly acetylation, in the nuclear import of histones H3 and H4. Our structural and biochemical observations demonstrate that Kap123 recognizes H3-NLS using two distally located lysine-binding pockets. In

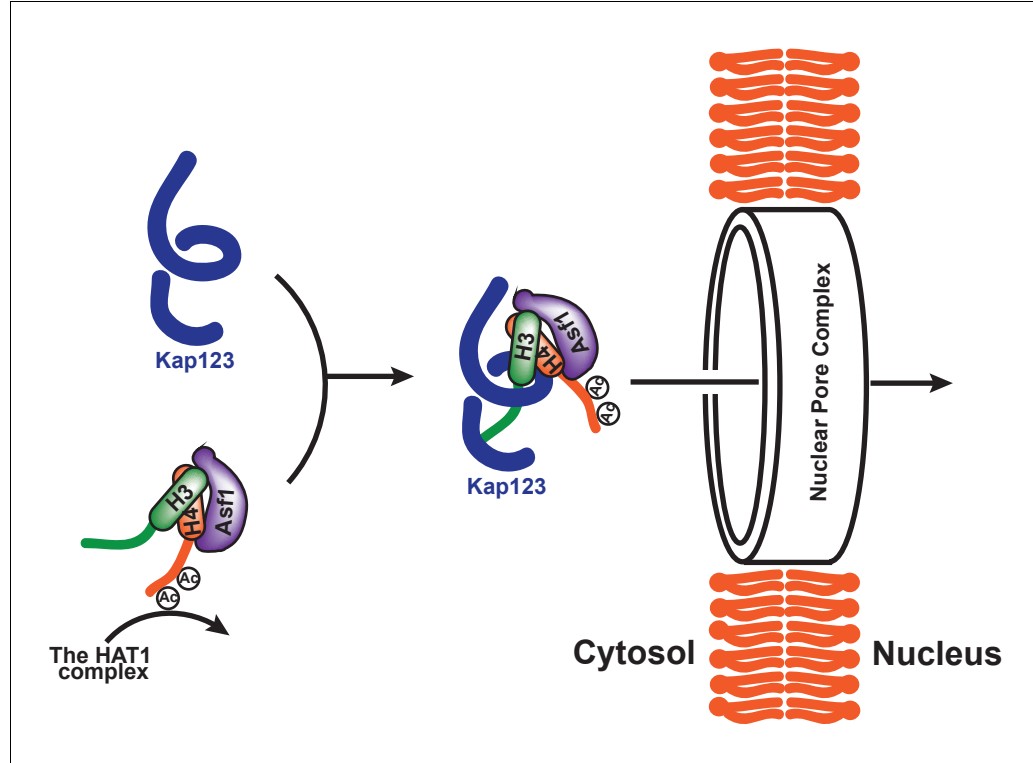

**Figure 7.** Proposed model of Kap123-dependent nuclear translocation of the H3:H4/Asf1 complex. Schematic model of the potential role of histone H4 diacetylation during nuclear import. Newly synthesized histones H3 and H4 are associated and immediately protected by its specific chaperone, Asf1. The HAT1 complex subsequently acetylates K5 and K12 of histone H4 as a part of the H3:H4/Asf1 complex. Diacetylation of the H4-NLS, whose affinity toward Kap123 is already fivefold weaker than H3-NLS, further destabilizes the Kap123-histone H4 interaction. Therefore, Kap123 preferentially associates with the H3-NLS and allows for histone H3-dependent Kap123 association during nuclear translocation. It should be noted that there are several histone H3 variants available in eukaryotes but there is only one known histone H4 protein, which can be commonly shared by each histone H3 variant.

DOI: https://doi.org/10.7554/eLife.30244.016

addition, the acetylation of key H3- and H4-NLS lysine residues negatively contributes to the Kap123-NLS association. Particularly, H4-NLS diacetylation may serve as an important step of the histone H3-dependent nuclear translocation of the H3:H4/Asf1 complex mediated by Kap123.

## Materials and methods

### Protein expression and purification

Full-length *Kluyveromyces lactis* (*Kl*) Kap123 (residues 1–1113) protein was cloned into the pET3a vector with $His_{X6}$ tag and tobacco etch virus (TEV) cleavage site at the N-terminus. The wild-type and mutant Kap123 proteins were expressed in the *escherichia Coli* Rosetta DE3 strain [RRID:WB-STRAIN:HT115(DE3)] with auto-inducible media (*Studier, 2005*). Harvested cells were resuspended and sonicated in Buffer A [30 mM Tris-HCl (pH 8.0), 500 mM NaCl, and 3 mM-mercaptoethanol] with protease inhibitors (PMSF, aprotinin, leupeptin, and pepstatin) and cell debris was eliminated by centrifugation. The cleared cell lysate was applied to the cobalt-affinity column (Qiagen, Hilden, Germany) pre-equilibrated with buffer A. Unbound proteins were washed out with buffer A and eluted with buffer A containing 200 mM imidazole. The Kap 123 N-terminal $His_{X6}$-tag was cleaved off with TEV protease (4°C, overnight incubation) in buffer B [30 mM Tris-HCl (pH 8.0), 100 mM NaCl, and 3 mM-mercaptoethanol] and further purified with additional cobalt columns to remove the N-terminal $His_{X6}$ tag. Flow-through fractions were then applied to HP-SP (GE Healthcare, Pittsburgh, PA)

connected with HP-Q (GE Healthcare) to remove contaminating proteins. Kap123 protein was eluted from HP-Q with increasing concentrations of NaCl. Fractions containing Kap123 were pooled, concentrated, and applied to the Superdex 200 size-exclusion chromatography (Prep grade 16/60, GE Healthcare) pre-equilibrated within 50 mM Tris-HCl (pH 8.0), 100 mM NaCl and 1 mM Tris (2-carboxylethyl)phosphine hydrochloride (TCEP). Kap123 was concentrated up to 10–15 mg/ml and used for crystal screening and optimization. Selenomethionine(Se-Met)-substituted Kap123 was expressed with PASM-5052 auto-inducible media and purified using the same procedure applied in native Kap123 purification (*Studier, 2005*).

## Crystallization of kl Kap123 alone and in complex with H3- and H4 NLS

Native and SeMet-substituted *Kl* Kap123 crystals were obtained using a hanging drop vapor diffusion method at 20°C by mixing with a reservoir solution of 0.1M sodium cacodylate (pH 6.5), 0.2M sodium acetate, 30% PEG 4000% and 5% Jeffamine M-600. Initial crystals appeared within one week and were used for microseeding to produce larger and better-diffracting crystals. The microseeding approach was further applied to obtain co-crystals of *Kl* Kap123 and H3-/H4-NLSs. The peptide sequences that we synthesized for this study were H3$_{1-28}$-NLS: $_1$ARTKQTARKSTGGKAPRKQLASKA ARK$_{-28}$ and H4$_{1-34}$-NLS: $_1$SGRGKGGKGLGKGGAKRHRKILRDNIQGITKPAI$_{-34}$. The crystals were cryoprotected in reservoir solution containing additional 15–20% glycerol and flash frozen in liquid nitrogen prior to data collection. All data were collected under cryogenic conditions (105 °K) at the beamlines 21ID-D and 21ID-G in LS-CAT (Advanced Photon Source, Argonne National Laboratory, USA).

## Data processing and structure determination of kl Kap123 alone and in complex with H3- or H4-NLS

A total of four single-wavelength anomalous diffraction (SAD) datasets from the SeMet-labeled brick-shaped crystals, in the space group P1, a = 79.05 Å, b = 88.12 Å, c = 102.01 Å, α = 79.19°, β = 80.03°, γ = 70.98°, were collected at the peak wavelength of selenium. The raw data sets were indexed, integrated, scaled and merged together by XDS through Xia2 (*Kabsch, 2010*). The crystallographic phase problem was resolved by the single-wavelength anomalous diffraction (SAD) method utilizing the anomalous scattering signal of Selenium from the SeMet-substituted crystal. Twenty-two selenium atoms (out of the total 24) from two copies of *Kl* Kap123 within an asymmetric unit were successfully located using SHELXC/D/E (*Sheldrick, 2010*), and the initial SAD phasing was calculated by PHENIX.autosol (*Terwilliger, 2000*). The initial model was manually built by utilizing selenium sites as a guidance using the program COOT (*Emsley et al., 2010*), and refinement calculations were carried out using the program Phenix.Refine (*Adams et al., 2010*). Native data sets for co-crystals of Kap123 and N-terminal peptides of histones H3 or H4 were integrated and scaled using HKL2000 (http://www.hkl-xray.com). Phases were calculated from the molecular replacement method using the apo *Kl* Kap123 structure as a search model. Molecular replacement computations were performed with Phaser (*McCoy et al., 2007*). Model building and structural refinement were performed using the same procedure as with the structure of *Kl* Kap123.

### Surface plasmon resonance analysis

A CM5 chip (GE healthcare) was coated with streptavidin in 10 mM acetate pH 4.5 at a flow rate of 5 μl/min. The biotinylated C-terminal histone H3 peptide (residues 1–35) was synthesized (Anygen, South Korea) and immobilized on the CM5 chip coated with streptavidin with a flow rate of 5 μl/min in the binding buffer (10 mM HEPES, 150 mM NaCl). Various concentrations of the wild-type and mutant *Kl* Kap123 protein (see figure legends of *Figures 3* and *4*) were prepared in the binding buffer with 2 mM TCEP and then injected at the flow rate of 30 μl/min in the binding buffer using a T200 instrument (GE Healthcare) at 7°C. After injecting the sample, the chip was regenerated by a regeneration buffer (1 M NaCl and 20 mM NaOH in binding buffer) at a flow rate of 30 μl/min. For measuring the interaction of wild-type *Kl* Kap123 and mutant histone H3 or H4 peptides, 0.1 μM Kap123 diluted in10mM acetate pH 4.0 was immobilized 7100 RU on the CM5 chip at flow rate of 5 μl/min. Peptides of histone H3 or H4 mutants were injected at a flow rate of 30 μl/min in the binding buffer and the binding was monitored at various concentrations (0, 0.31, 0.62, 1.25, 2.5, 5 and 10 uM) at 7°C. For the H4K16Q peptide, Kap 123 was immobilized with 5100 RU on CM5 chip, and

H4K16Q and wild-type H4 peptides (0, 0.7, 1.5, 3, 6, 12 and 25 uM) were injected as analytes at a flow rate of 30 µl/min at 7°C.

## Competition assay of H3- and H4-NLSs toward Kap123 using pull-down

MBP-tagged Kap123 was preincubated with either C-terminal Sumo-tagged histone $H3_{1-59}$ or $H4_{1-48}$ in buffer B [30 mM Tris–HCl (pH 8.0), 50 mM NaCl, 1 mM DTT, and 0.01% NP-40 (w/v)] for 15 mins. The binding reactions were challenged by the increased amount of competitors, $H3_{1-59}$-NLS-Sumo to compete with $H4_{1-48}$-NLS-Sumo, and $H4_{1-48}$-NLS-Sumo or $H4_{1-48}$-NLS$^{K5QK8QK12Q}$-Sumo to compete with $H3_{1-59}$-NLS-Sumo. The reactions were then incubated in the presence of amylose beads (New England Biolabs, Ipswich, MI) pre-equilibrated with buffer B for 90 mins at 4°C. The beads were washed three times with buffer B, and bead-bound proteins were subjected to denaturing polyacrylamide gel electrophoresis.

## Acknowledgements

We thank the staff at the Advanced Photon Source LS-CAT beamlines for their advice and assistance with data collection. Ms. Jennifer Chik for help with the manuscript preparation. This work was supported in part by grants (1–16-JDF-017, N019154-00, AG050903 and DK111465) to U-SC, and by grants (NRF-2016R1A2B3006293, NRF-2013R1A1A2055605, NRF-2014K2A3A1000137 and 2011–0031955) to JS.

## Additional information

### Funding

| Funder | Grant reference number | Author |
|---|---|---|
| National Institutes of Health | DK111465 | Uhn-soo Cho |
| American Diabetes Association | 1-16-JDF-017 | Uhn-soo Cho |
| National Institutes of Health | AG050903 | Uhn-soo Cho |
| March of Dimes Foundation | N019154-00 | Uhn-soo Cho |
| National Research Foundation of Korea | 2016R1A2B3006293 | Ji-Joon Song |
| National Research Foundation of Korea | 2013R1A1A2055605 | Ji-Joon Song |
| National Research Foundation of Korea | 2014K2A3A1000137 | Ji-Joon Song |
| National Research Foundation of Korea | 2011-0031955 | Ji-Joon Song |

The funders had no role in study design, data collection and interpretation, or the decision to submit the work for publication.

### Author contributions

Sojin An, Conceptualization, Resources, Data curation, Formal analysis, Validation, Visualization, Methodology, Writing—original draft; Jungmin Yoon, Data curation, Formal analysis, Validation, Visualization, Writing—review and editing; Hanseong Kim, Data curation, Formal analysis, Validation, Writing—review and editing; Ji-Joon Song, Data curation, Supervision, Funding acquisition, Writing—review and editing; Uhn-soo Cho, Conceptualization, Resources, Formal analysis, Supervision, Funding acquisition, Validation, Investigation, Visualization, Writing—original draft, Project administration, Writing—review and editing

### Author ORCIDs

Uhn-soo Cho (iD) http://orcid.org/0000-0002-6992-2455

Decision letter and Author response
Decision letter https://doi.org/10.7554/eLife.30244.025
Author response https://doi.org/10.7554/eLife.30244.026

## Additional files

### Supplementary files
• Supplementary file 1. Data collection and refinement statistics.
DOI: https://doi.org/10.7554/eLife.30244.017

• Transparent reporting form
DOI: https://doi.org/10.7554/eLife.30244.018

### Major datasets
The following datasets were generated:

| Author(s) | Year | Dataset title | Dataset URL | Database, license, and accessibility information |
|---|---|---|---|---|
| An S, Kim H, Cho US | 2017 | Crystal structure of full-length Kluyveromyces lactis Kap123 | https://www.rcsb.org/pdb/explore/explore.do?structureId=5VCH | Publicly available at the RCSB Protein Data Bank (accession no: 5VCH) |
| An S, Kim H, Cho US | 2017 | Crystal structure of full-length Kluyveromyces lactis Kap123 with histone H3 1-28 | https://www.rcsb.org/pdb/explore/explore.do?structureId=5VE8 | Publicly available at the RCSB Protein Data Bank (accession no: 5VE8) |
| An S, Kim H, Cho US | 2017 | Crystal structure of full-length Kluyveromyces lactis Kap123 with histone H4 1-34 | https://www.rcsb.org/pdb/explore/explore.do?structureId=5W0V | Publicly available at the RCSB Protein Data Bank (accession no: 5W0V) |

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
