## [Decision Letter]

Thank you for submitting your article "Structure-based nuclear import mechanism of histones H3 and H4 mediated by Kap123" for consideration by *eLife*. Your article has been reviewed by 3 peer reviewers, and the evaluation has been overseen by Eric Campos as the Reviewing Editor (who served as one of the reviewers) and Jessica Tyler as the Senior Editor.

The reviewers have discussed the reviews with one another and the Reviewing Editor has drafted this decision to help you prepare a revised submission.

Summary:

These are interesting findings providing the crystal structures of the *K. lactis* Kap123 karyopherin protein by itself, with histone H3, and with histone H4. Kap123 is shown to have a unique binding mechanism that uses two lysine-binding pockets to bind histone NLS. Kap123 preferentially binds H3-NLS over H4-NLS, and histone acetyl-lysine mimics (glutamine substitutions) further weaken H4 binding.

The study provides molecular details into the nuclear import of newly synthesized histones, the findings of which will surely interest labs studying chromatin biology and the broader scientific community. We recommend that the manuscript be published, but the authors should incorporate some modifications as per the points below.

Essential revisions:

1) Carefully distinguish the effect of H4 K5/K8/K12 binding to Kap123 from that of H4 K16 in the text. This is important since the Kap123-H4-NLS structure shows that the second lysine-binding pocket binds K16 of H4-NLS, yet direct Kap123 binding to H4 K5/K12 is a real possibility since it is the acetylation of these residues that affects Kap123 binding to H4.

2) Comment (in the Discussion) on the difference between the histone constructs used in this study in comparison to past studies, and how there may be additional contact residues in the histone globular domain to which karyopherins may also bind.

3) Models where cytoplasmic acetylation may modulate binding within an import complex where there are at least 2 NLSs, and/or that acetylation may help dissociate the complex more readily once in the nucleus are valid. However, please be sure to refer to all published data. Right now the text only discusses the observation that H4K5Q/K8Q/K12Q diminishes H4-NLS nuclear import in yeast (e.g. JBC 282:20142-50), but does not discuss results in other systems suggesting that H4 acetylation is important for nuclear import (e.g. JBC 286: 17714-21; MBC 22:245-55).

The authors should also not just focus on H4 in their Discussion and include the possibility that both the H3 NLS and H4 NLS may be acetylated to modulate binding.

4) Remove the subtitle 'Acetylation of key lysine residues at H3 1-28 -NLS interferes with Kap123 association', since no experiments are shown using acetyl H3 in this subsection.

5) The structural data is solid on its own. We however encourage the authors to consider obtaining the following experimental results prior to resubmission (as feasible):

a) Include Biacore experiments for H4K16Q and of H3K14K23A (since the H4K16A and H3K14QK23Q are shown). The data would also be valuable considering the structural data.

b) Using a H4K5Q/K12Q protein instead of the K5Q/K8Q/K12Q construct in the competition assay, for consistency with the Biacore experiments and since K5 and K12 are seen as the main pre-deposition acetyl marks.

---

## [Author Response]

1) Carefully distinguish the effect of H4 K5/K8/K12 binding to Kap123 from that of H4 K16 in the text. This is important since the Kap123-H4-NLS structure shows that the second lysine-binding pocket binds K16 of H4-NLS, yet direct Kap123 binding to H4 K5/K12 is a real possibility since it is the acetylation of these residues that affects Kap123 binding to H4.

We agree with the reviewers’ point. Our SPR data also indicate that H4 K16 mutations are less significant than H4 K5/K12 mutations in Kap123 association. As we stated in the manuscript, the mutational effect of H4 K16 might be less significant due to additional contacts mediated by H4 R17 and R19. We expect that the H4 K16 interaction with the second lysine-binding pocket is specific but rather its affinity is relatively weak. However, H4 K5 and K12 may contribute more in Kap123 association based on SPR and competition assays. We updated the revised manuscript accordingly (subsection “Acetylation mimic mutations of H3- and H4-NLSs reduce the affinity toward Kap123” and subsection “H4_1–34_-NLS mainly interacts with Kap123 via the second lysine-binding pocket”).

2) Comment (in the Discussion) on the difference between the histone constructs used in this study in comparison to past studies, and how there may be additional contact residues in the histone globular domain to which karyopherins may also bind.

As the reviewers pointed out, we designed and used peptides of histone H3- and H4-NLSs (H3_1–28_ and H4_1–34_) for our crystallographic studies based on the former publication (JBC, 2007, 282:20142-50). However, we used different sizes of H3- and H4-NLSs for SPR experiment (H3_1–35_ and H4_1–21_) and the competition assay (H3_1–59_ and H4_1–48_) partly based on crystal structures we determined and the former publication (JBC, 2002, 277:862-868), respectively. Particularly, H4_1-21_ used for the SPR experiment may not fully restore the interaction with Kap123. Mosammaparast et al. (JBC, 2002, 277:862-868) suggested that the additional H4-Kap123 interaction might exist through H4_22–42_. Although we don’t see an additional interaction of H4_22–34_ in our crystal structure (Kap123-H4_1–34_-NLS), additional interaction may exist in H4_35–42_ or the electron density of additional interaction via H4_22–34_ was not clearly visible like as the case of histone H4 K5, K8, and K12. We made additional comments in the Discussion accordingly (subsection “Acetylation mimic mutations of H3- and H4-NLSs reduce the affinity toward Kap123”).

3) Models where cytoplasmic acetylation may modulate binding within an import complex where there are at least 2 NLSs, and/or that acetylation may help dissociate the complex more readily once in the nucleus are valid. However, please be sure to refer to all published data. Right now the text only discusses the observation that H4K5Q/K8Q/K12Q diminishes H4-NLS nuclear import in yeast (e.g. JBC 282:20142-50), but does not discuss results in other systems suggesting that H4 acetylation is important for nuclear import (e.g. JBC 286: 17714-21; MBC 22:245-55).The authors should also not just focus on H4 in their Discussion and include the possibility that both the H3 NLS and H4 NLS may be acetylated to modulate binding.

Some papers came up with the different conclusion although our data still agree well with most of previous biochemical studies [Barman et al., (2006) BBRC, 345, 1547-1557; Ma et al., (1998) PNAS 95, 6693-6698; Mosammaparast et al., (2002) JBC 277, 862-868; Blackwell et al., (2007) JBC, 282, 20142-20150; Soniat et al., (2016) JBC 291, 21171-21183; Glowczewski et al., (2004) MCB 24, 10180-10192; Agudelo Garcia et al., (2017) Nuc. Acid Res. 45, 9319-9335]. Ejlassi-Lassallette et al. (MBC 22:245-55) showed that exogenously introduced histones H3:H4^K5Q/K12Q^ improved the nuclear import in *P. polycepthalum*. Although their result is intriguing, we cannot directly compare their results to ours because several karyopherin proteins in addition to Kap123 homolog may participate in nuclear transportation of histones H3:H4 in vivo. Highly charged Flag tag at the N-terminal of histone H4 (DYKDDDDK epitope) may also contribute in nuclear import process. They also proposed that Hat1 participates in the nuclear transport of histones H3:H4, but it is widely known that Hat1 deletion does not have a significant phenotype and Imp4 associated histones H3:H4 contains Asf1a and Asf1b but not Hat1 (JBC 286:117714-21). Alvarez et al. (JBC 286:117714-21) performed the pull-down assay of Imp4 using pre-incubated with either wild-type H4_2-20_ or tetra-acetylated H4_2-20_^K5/K8/K12/K16Ac^ peptides and subsequent addition of histone (H3:H4)_2_ tetramer for the competition. However, our crystal structure, SPR, and the competition assays all clearly show that acetylation of histone H4 K5/K8/K12/K16 weakens the Kap123 interaction. They further showed that histone H4-NLS^K5Q/K12Q^ –EYFP improved nuclear import in vitro nuclear transportation assay using HeLa cell reticulocyte extract. However, this result does not agree with other earlier observation (JBC 282:20142-50), in which H4K5Q/K8Q/K12Q diminishes H4-NLS nuclear import in yeast. However, we agree on the reviewer’s point that we should include in the Discussion the potential effect of histone H3 acetylation. Our crystal structure and binding results clearly indicate that acetylation of histone H3 K14 and K23 significantly influences to the Kap123-H3 interaction and indeed H3 K14 acetylation has been described in former literatures although results vary in different species and different studies. We discussed this possibility in the revised manuscript (subsection “Diacetylation of H4-NLS may allow Kap123-H3 dependent nuclear translocation of the H3:H4/Asf1 complex”).

4) Remove the subtitle 'Acetylation of key lysine residues at H3 1-28 -NLS interferes with Kap123 association', since no experiments are shown using acetyl H3 in this subsection.

Accordingly, we deleted the subtitle.

5) The structural data is solid on its own. We however encourage the authors to consider obtaining the following experimental results prior to resubmission (as feasible):a) Include Biacore experiments for H4K16Q and of H3K14K23A (since the H4K16A and H3K14QK23Q are shown). The data would also be valuable considering the structural data.

We now include Biocore results of H4K16Q and H3K14A/K23A in Figure 5 and updated the graph (Figure 5) accordingly.

b) Using a H4K5Q/K12Q protein instead of the K5Q/K8Q/K12Q construct in the competition assay, for consistency with the Biacore experiments and since K5 and K12 are seen as the main pre-deposition acetyl marks.

As the reviewers suggested, we performed the competition assay using histone H4 K5Q/K12Q mutant and we confirmed that K5Q/K12Q mutation still barely competes with histone H3 tail. We updated Figure 6 accordingly.